# Intramuscular Injections and Dry Needling within Masticatory Muscles in Management of Myofascial Pain. Systematic Review of Clinical Trials

**DOI:** 10.3390/ijerph18189552

**Published:** 2021-09-10

**Authors:** Zuzanna Nowak, Maciej Chęciński, Aleksandra Nitecka-Buchta, Sylwia Bulanda, Danuta Ilczuk-Rypuła, Lidia Postek-Stefańska, Stefan Baron

**Affiliations:** 1Department of Temporomandibular Disorders, Medical University of Silesia in Katowice, Traugutta sq. 2, 41-800 Zabrze, Poland; aleksandra.nitecka@sum.edu.pl (A.N.-B.); sbaron@sum.edu.pl (S.B.); 2Preventive Medicine Center, Komorowskiego 12, 30-106 Kraków, Poland; maciej@checinscy.pl; 3Department of Pediatric Dentistry, Medical University of Silesia in Katowice, Traugutta sq. 2, 41-800 Zabrze, Poland; bulanda.sylwia@gmail.com (S.B.); danuta.ilczuk@gmail.com (D.I.-R.); swrzab@sum.edu.pl (L.P.-S.)

**Keywords:** myofascial pain, temporomandibular disorders, intramuscular injections, dry needling, wet needling, Botox

## Abstract

Background: Myofascial pain is an important cause of disability among the whole population, and it is a common symptom of temporomandibular joint disorders (TMDs). Its management techniques vary widely; however, in recent years, there has been a growing interest especially in needling therapies within masticatory muscles, due to their simplicity and effectiveness in pain reduction. Methods: The construction of the following study is based on PICOS and PRISMA protocols. A systematic literature search was conducted based on the PubMed and BASE search engines. Searching the abovementioned databases yielded a total of 367 articles. The screening procedure and analysis of full texts resulted in the inclusion of 28 articles for detailed analysis. Results: According to analyzed data, clinicians manage myofascial pain either with wet or dry needling therapies. The most thoroughly studied approach that prevails significantly within the clinical trials is injecting the botulinum toxin into the masseter and temporalis. Other common methods are the application of local anesthetics or dry needling; however, we notice the introduction of entirely new substances, such as platelet-rich plasma or collagen. In the analyzed articles, the target muscles for the needling therapies are most commonly localized by manual palpation although there are a variety of navigational support systems described: EMG, MRI or EIP electrotherapy equipment, which often aid the access to located deeper lateral and medial pterygoid muscle. Conclusions: Needling therapies within masticatory muscles provide satisfactory effects while being simple, safe and accessible procedures although there still is a need for high quality clinical trials investigating especially injections of non-Botox substances and needling within lateral and medial pterygoid muscles.

## 1. Introduction

Myofascial pain is an important cause of disability among men and women of all ages and may affect up to 85% of the population [1,2]. It is a common symptom of temporomandibular disorder (TMD) [3]. Its etiology is vast and complex, and often requires a multidisciplinary approach not only during treatment, but also during the diagnostic process [4]. The Diagnostic Criteria for Temporomandibular Disorders (DC/TMD) places myofascial pain within its Taxonomic Classification under the II category among other masticatory muscles disorders [5]. The 2020 International Classification of Orofacial Pain (ICOP) allows for a more systematized and standardized look at myofascial pain [6]. It specifies definitions and diagnostic criteria for primary myofascial orofacial pain: acute and chronic, as well as secondary myofascial orofacial pain attributed to tendonitis, myositis and muscle spasm. The first edition of ICOP by the International Headache Society is a clear diagnostic classification aligned to the International Classification of Headache Disorders, 3rd edition (ICHD-3), DC/TMD and International Classification of Diseases 11th Revision (ICD-11), which makes it a comprehensive tool in proper myofascial pain diagnosis [6].

Pathologies within masticatory muscles manifest with pain and functional disorders. They are caused by excessive muscle effort, leading to muscle damage. High muscle activity is most often associated with parafunctional activities within the masticatory system, such as bruxism or increased emotional tension related to severe stress or depression [7,8]. The initial contracture of the sarcomeres leads to decreased capillary blood circulation and increased anerobic metabolism. Emerging symptoms are painful myofascial trigger points (MTPs) within the muscles. Trigger points are areas of local hypoxia, ischemia, inflammation and neurophysiologic changes at nociceptors. They lead to peripheral and central pain sensitization and may cause the appearance of deep referred pain, deep to and distant from the initial stimulus, due to sensitization of peripheral nociceptors, spinal dorsal horn neurons and the brainstem [1,9,10]. However, in patients with chronic pain conditions, central nervous system sensitization has already occurred. Therefore, in those cases especially, it remains to be elucidated whether the appearance of MTPs may be considered a consequence of that sensitization or a contribution to that process [10]. The most widespread definition of MTPs was suggested in 2019 by Simons at al.: myofascial MTP is a hyperirritable spot within a taut band of skeletal muscle that is painful on compression, stretch, overload, or contraction of the tissue, which usually responds with referred pain that is perceived distant from the spot [11]. MTPs can be either active or latent. Their presence weakens the muscle, can prevent muscle lengthening, and increases co-antagonist activation, hence the accompanying symptoms, such as limited mouth movements. [10,12].

Myofascial pain treatments vary from conservative methods, including exercises, physiotherapy (manual therapy and transcutaneous electrical nerve stimulation, low level laser therapy), biofeedback, occlusal appliance therapy or pharmacotherapy (muscle relaxants, tricyclic antidepressants), to invasive, including dry needling, wet needling with the use of various substances or even acupuncture [1]. In many clinical trials, needling methods were used when all available conservative treatments failed or as a supporting treatment [13,14].

In recent years, there has been a growing interest in the invasive methods of managing myofascial pain, due to their effectiveness and simplicity [10]. In 2020, Al-Moraissi et al. published a meta-analysis of randomized clinical trials comparing treatment outcomes of different needling therapies [15]. The authors concluded significant improvement in maximum mouth opening, decreasing pain levels and an increasing pressure pain threshold as the outcome of the specific needling therapies when compared to placebo or acupuncture. Although their results raise a serious question regarding the mechanism of pain reduction following the needling therapies, the authors stated that the effectiveness of needling therapy did not depend on the needling type (dry or wet) nor on the needling substance. Al-Moraissi et al. also highlighted the problem of low-quality evidence due to imprecisions and inconsistencies in the techniques applied in analyzed clinical trials [15]. Therefore, there is a justified need for further high-quality research and its compilation in the form of systematic reviews.

The aim of the following systematic review was to compare and summarize the current approaches and techniques implemented in needling therapies of masticatory muscles. The analysis of the literature provokes a few statements. Primarily, there is high inconsistency in the currently applied treatment methods; therefore, their objective comparison is very difficult. Secondly, there is a need for more clinical trials concerning the lateral and medial pterygoid muscles as well as trials investigating non-Botox treatments. Moreover, a standardized needling therapy protocol could be useful in further clinical trials.

## 2. Materials and Methods

### 2.1. Protocol Registration

The protocol of this systematic review was developed on 20 May 2021 and submitted to the scientific council of the Medical University of Silesia in Katowice, Poland. It was registered as a part of scientific research of SUM number KNW-1-085/N/9/I with the Bioethical Committee PCN/0022/KB1/95/I/19. The systematic review was registered in the PROSPERO database with ID number 272124. This systematic review was conducted in accordance with the Preferred Reporting Items for Systematic Reviews and Meta-Analyzes (PRISMA) statement (Appendix A). In the course of the systematic review, eligibility criteria were established and respected, using the PICOS framework [16].

### 2.2. Data Sources and Search Strategy

The MEDLINE and PubMed Central (PMC) as well as other databases, including journal archives, were searched, using the PubMed search engine and BASE, i.e., Bielefeld Academic Search Engine, on 18 April 2021 [17,18]. At the time of the search, the PubMed engine serviced databases with a total of over 32 million records, and BASE, databases with over 240 million records [17,18]. The structure of each search strategy is presented in Table 1.

### 2.3. Eligibility Criteria

Eligibility criteria for this systematic review were developed according to the PICOS framework [16]. These were used to develop a search strategy in the medical databases of records and the subsequent stages of article selection. In line with the PICOS acronym, positive (inclusion) and negative (exclusion) criteria were adopted. (1) In line with the “Population” trait, studies on groups of adult patients suffering from myofascial pain in the mouth and/or face were included. Studies on animals and studies on human patients suffering from specific diseases and syndromes listed in the appropriate cell of Table 2 were excluded. Studies in which pain was induced by researchers were also rejected, due to the additional injection necessary for that purpose. (2) Therapeutic punctures into the masticatory muscles, including both dry needling and the administration of any substances, were considered necessary for the qualification of the article in the context of “Intervention”. However, punctures which, apart from the masticatory muscles, also affected other muscles, served nerve blocks, or served acupuncture, were rejected. (3) No “Comparison” was required, i.e., research work was accepted regardless of the type of control group or its absence. (4) With regard to “Outcome”, the description of the puncture technique allowing reaching the interior of any masticatory muscle was adopted as a necessary condition. Additionally, it was required to indicate in the article any of the positive puncture effects mentioned in the appropriate cell of Table 2. (5) As a positive criterion, “Study design” clinical studies were required with an evidence level of 1 to 3, i.e., randomized controlled trials, prospective cohort studies and retrospective case control studies were accepted. Articles published before 2011 and not available in English were rejected. The full PICOS criteria, i.e., with additional details, are presented in Table 2 [16].

### 2.4. Study Selection

The following systematic review was conducted based on the Preferred Reporting Items for Systematic Reviews and Meta-Analyses (PRISMA) guidelines, with the workflow presented in Figure 1 [19]. Initially, all identified records were synthesized using the Rayyan QCRI application [20], the duplicates were removed, and afterward, the titles and abstracts were blindly screened by two of the authors (Z.N. and M.C.) according to the PICOS criteria. Cohen’s Kappa coefficient was used to determine the convergence of the assessments in the screening phase, and the result was κ = 0,98. Each time a decision conflict of whether to include or exclude an article occurred, the abstract was qualified to the next evaluation level—full-text assessment. Afterward, three of the authors (Z.N., M.C. and A.N.-B.) blindly evaluated the full-texts, reaching total convergence of the results. The detailed risk assessment does not apply in our work, as in this review, we did not intended to assess the effectiveness of the needling therapies but to evaluate the similarities and divergences.

### 2.5. Data Extraction

The data were extracted by three authors (Z.N., M.C. and A.N.-B.), who obtained the following information from the content of the articles: (1) author, year of publication; (2) study group size; (3) wet/dry needling (injected substance); (4) navigation method or support system; (5) needle penetration spot(s); (6) amount of injected substance (per muscle); and (7) number of treatment sessions. These data were compiled in a table using the Google Sheets software (Google LLC, Mountain View, CA, United States).

## 3. Results

In this systematic review, the authors included 28 scientific papers that described with sufficient detail the techniques of intramuscular needling (dry needling and injections) into masticatory muscles. The extracted data were tabulated, using the main breakdown depending on the muscle that was the target of the intervention. Some of the studies compared multiple treatment methods or administration of various substances, so every time different methods of injections, substances used, or muscles treated were described, the authors decided to analyze those as separate trials. Thus, four tables were created corresponding to the following muscles: masseter, temporal, lateral and medial pterygoid (Table 3, Table 4, Table 5 and Table 6). The following data were extracted from the selected papers: author and year of publication, study population sample size, method of needling applied in the study, navigation method or additional support system used, needle penetration spot, amount of injected substance per muscle and number of treatment sessions. The abbreviations used in Table 3, Table 4, Table 5 and Table 6 are as follows: DDN—deep dry needling; SDN—superficial dry needling; BTX—Botox; LA—local anesthesia; PRP—platelet-rich plasma; EMG—electromyography; PNE—percutaneous needle electrolysis; ns—not specified.

As interest in needling therapies for masticatory muscles and MTPs has been gradually growing throughout the recent years, there has been an increase in the number of reviews, meta-analysis as well as clinical trials referring to the subject, although no specific expository guidelines have been set. In the material we analyzed, 24 out of 28 articles (86%) were published in 2016 and later, i.e., in the second half of the period we analyzed. Some papers described needling of more than one muscle, and in this case, the publication appears in our review many times, separately for each described muscle and for the administered substance. We recorded thirty-two trials on masseter needling, making this muscle the most frequently described muscle in our data. Thirteen studies looked at the temporal muscles, five papers dealt with needling of the lateral pterygoid muscles, and only one paper looked at the medial pterygoid muscle.

The cited studies differed significantly in the size of the studied groups. For the entire material, the minimum size of the study group was 5, and the maximum size was 61. For the latter, the most numerous study group was injected with botulinum toxin into the masseter muscles. Among the articles describing the needling of the masseter muscles, 20 were based on a group of patients equal to or greater than 20. Nine studies on temporal muscles and four studies on lateral pterygoids met the same condition. The total number of patients’ cases for the masseter, temporal, lateral and medial muscles was 711, 294, 90 and 5 cases, respectively. Analyzing the summary data, it is apparent that the masticatory muscle most commonly treated by needling interventions is the masseter (65%), followed by the temporal muscle (27%). Subsequently, with a clearly smaller number of interventions, rank the lateral pterygoid muscle (8%) and medial pterygoid muscle (less than 1%). The data are presented in Figure 2.

Taking into account the substances injected into the masticatory muscles in general, the effects of botulinum toxin injections were studied most frequently. For the masseter muscles, 17 studies on botox were carried out: 7 involving various local anesthetics, 4 with deep dry needling, 2 with platelet-rich plasma, 1 with surface dry needling and 1 with the injection of collagen (Figure 3). All studies of temporal muscle needling looked at the administration of Botox, except one, where lidocaine was administered. Pterygoid muscle needling has been scarcely studied and techniques have not been standardized. In two studies, Botox was administered to the lateral pterygoid muscles; in two others, deep dry needling of these muscles was performed; and in one study, percutaneous needle electrolysis was used. The study on the group of five patients on the medial pterygoid muscle was aimed at investigating and describing the technique of deep dry needling.

Most authors indicated palpation as a method of locating anatomical points and MTPs and thus, navigation during needling techniques (72.5%). Further methods of establishing the puncture target spot require various navigational aids: guidance systems, such as EPI ^®^ electrotherapy equipment (3.9%); EMG (11.8%) or the use of imaging methods, such as MRI (2.0%). The above-mentioned methods are described in detail in the discussion section of this paper. In the remaining 9.8% of the trials, the method of determining the point of puncture was not specified (Figure 4).

The number of puncture sites seems to depend on the individual decision of the operators. More than half of the trials set out a specific number of injection sites per muscle treated (51%). An alternative approach applies, setting the MTPs as the direct injection spots (37%), which results in a further need to identify them within the muscles. A few included trials lacked the information about the desired needle penetration site (12%). Regardless of the type of needling therapy, the techniques requiring two to three injections dominated. Single punctures are not commonly practiced and were indicated by only three authors administering Botox or an anesthetic. On the other hand, Guarda-Nardini (2012) described the administration of Botox with numerous punctures arranged in the shape of a chessboard.

In the case of the masseter muscles, Botox dosing was the most frequent. The authors used here from 20 UI up to 150 UI. Doses from 30 to 50 UI dominated. In the case of local anesthetics, 0.2 to 0.5 mL of anesthetic fluid was used per puncture, and Nitecka-Buchta et al. used 2 ml of LA per muscle treated. With regard to temporal muscles, the doses of botulinum toxin were lower than in the case of masseter muscles and ranged from 10 to 100 IU, with the dominant values from 20 to 25 IU. For the only study reporting the administration of lidocaine to the temporal muscles, dosing was not specified. Two studies of Botox administration in the lateral pterygoid muscles indicated administration of 10 and 20 IU.

## 4. Discussion

Upon the explored studies, it seems most appropriate to distinguish two elementary techniques of needling: wet needling (injections) and dry needling. Apart from that, another major difference in approaches points to highlighting needling therapies based on the manual localization of the target muscle and appropriate puncture spots as well as ones that rely on additional navigation methods or support systems. As a detailed description of all the techniques applied in analyzed trials reaches beyond the scope of this review, we focused on highlighting controversial and diverging hypotheses as well as presenting the current state of the research in the field.

### 4.1. Intramuscular Injections (Wet Needling)

Wet needling refers to interventions based on introducing a particular substance into the muscle tissue, whereas dry needling refers to insertion of a solid filiform needle into the muscle without administering any substances. Discussing wet needling, most of the researchers seem to lean toward well-established treatment methods, such as injection of Botox or local anesthetics although in recent years, there has been an ongoing search for another substance that would turn out to be more beneficial in myofascial pain therapy. Since 2018, novel clinical trials started emerging in the field, and three of them are included in this review. Two of the papers describe injections of PRP, and one deals with collagen administration. PRP injections allow for reintroduction of the main growth factors from native blood in a concentrated form directly into the damaged muscle, which promotes its healing. Growth factors, such as insulin-like growth factor, fibroblast growth factor, transforming growth factor beta, hepatocyte growth factor, tumor necrosis factor, platelet-derived growth factor and prostaglandins support muscle regeneration and myogenesis by stimulating the proliferation and differentiation of myoblasts. PRP can increase collagen and extracellular matrix production as well as reduce inflammation [32,46,47]. Multiple studies have also shown that PRP administration reduces the regeneration time and improves functional recovery of the skeletal muscles [48,49]. Al-Moraissi et al. in his network meta-analysis based on 21 clinical trials (959 patients) stated that PRP administration was the most effective treatment with short-term post-treatment (1–20 days) pain reduction [15]. Taking into account that PRP stimulates physiological processes within the organism, successfully aids myofascial pain treatment and, at the same time, carries no risk of any serious adverse effects, it seems fair to recommend its extended use in future clinical trials, especially as for now, the available research presents mostly low-quality evidence [46]. Collagen injections similarly promote natural regeneration processes within muscles. Delivery of the collagen supports development of the muscle cells, and it is a vital surrounding cell that coordinates cell behavior and communication. It contributes to decreasing apoptosis and increasing myoblasts proliferation. Collagen is also present in the nervous system and participates in proper nerve myelination. It has been proved that muscle injuries are followed by a few days of elevated collagen production [9,50]. Included in this review, a clinical trial by Nitecka-Buchta et al. comparing injections of collagen and lidocaine showed promising results, proving the superiority of collagen administration although further long-term clinical trials on larger samples are still needed [9]. Botulinum toxin, being the most commonly administered drug in myofascial pain treatment, shows significantly lower effectiveness than LA and slightly lower effectiveness than dry needling intermediate-term, according to Al-Moraissi [15]. Moreover, there are reports claiming the presence of adverse effects of multiple BTX injections into masticatory muscles. Raphael et al. noted decreased bone density in all the trial participants exposed to BTX and none in the unexposed ones [51]. However, their study is characterized by very low evidence levels, due to an insufficient study group size. In 2020, Canales et al. showed a significant decrease of coronoid and condylar processes’ bone volume that was related to doses of botulinum toxin [52]. In the medium-dose group, they observed bone loss of the coronoid process alone, while in the high dose group, both coronoid and condylar processes were affected. However, it is worth highlighting that application of low doses of BTX did not cause any bone-related damage and is a treatment recommended by the authors [52]. On the contrary, in 2020, Raphael et al., based on a cohort consisting of 35 patients, stated that multiple Botox injections failed to produce clinically significant bone changes [53]. Nevertheless, those studies indicate that multiple BTX treatments might be a cause of some significant adverse effects, and there is a need to monitor the patients undergoing BTX therapy, especially when high doses of the toxin are applied. More studies on larger samples investigating the problem are needed to objectively assess the real scale of the risk.

### 4.2. Dry Needling

Dry needling as described by The Australian Society of Acupuncture Physiotherapists is as follows: rapid, short-term needling to altered or dysfunctional tissues in order to improve or restore function; it may be performed with acupuncture needle or any needle without the injection of the fluid [54]. Presumably, the most common dry needling approach was suggested by Hong and it concerns the “fast in, fast out” method that requires obtaining the local twitch response (LTR) to confirm the proper placement of the needle penetrating the MTP [55]. However, recent studies by Hakim et al. as well as the review by Perreault et al. suggest that eliciting LTR is not necessary for successfully managing myofascial pain [56,57]. Moreover, Hakim et al. implied the superiority of a few sessions of dry needling without obtaining LTR in long-term clinical observation, presumably due to the fact that repeated sessions of dry needling may cause increased connective tissues’ damage and impair the reinnervation process, which is a greater concern when applied to more aggressive methods requiring obtaining LTR [56]. Nevertheless, concluding from the analyzed articles, LTR may be a useful confirmation of proper needle placement within MTPs, especially during needling of LPM and MPM without the aid of navigational systems. Among the included trials discussing dry needling, alternative techniques were also described. Ozden et al. compared deep dry needling and superficial dry needling [33]. They adopted an acupuncture-like technique and introduced the needle into muscle tissue up to 10 or 5 mm, respectively; then, after immersion, the needle was rotated clockwise, left for 10 min and then rotated again. Interestingly, the abovementioned clinical trial conducted on 40 patients implies better pain-reduction efficacy of the superficial dry needling approach.

### 4.3. Techniques Based on Manual Localization

Within studies describing the manual method, determining the puncture spot relies on the palpation of the muscles’ bellies either manually or digitally with the use of a digital algometer. It allows for defining the localization of MTPs and evaluation of the accurate reach of the muscles. Injecting and dry needling directly in the MTPs is a common approach and seems most reasonable. The advantage of this method is operating straight on the pain causal spot, which allows for direct reduction of the excitability of the central nervous system by reducing peripheral nociception from the trigger point [10]. However, in other studies, the puncture target spots were marked in a standardized manner, uniform for all the study participants with the anatomical landmarks adopted as the guidelines. Approaches suggested in the analyzed material include injections as follows: across 2 cm skin surface over the most prominent area of the muscle after clenching [27]; regularly across the muscle mass [42]; into area over the greatest cross-section surface of both masseter bellies [35]; regularly along the long axis of the masseter muscle [26]; or regularly in the lower part of the muscle [21]. Although simple, those methods are only applicable for easily accessible muscles—temporal and masseter. Decisions about the site localizations are based mostly on the anatomy of the particular muscle. Most of the analyzed clinical trials described a specific number of injection sites per muscle, although it differed from 2 to 5 for masseter and from 2 to 3 for temporalis. This discrepancy is mainly due to the difference in volume, mass and surface of these muscles. Placing a few injection spots, usually about 1 cm apart, also finds its justification. The administered substances have the ability to infiltrate the surrounding tissues. According to Borodic et al., Botox can spread for about 30–40 mm from the puncture spot, and according to Kim et al., to 10 mm [14,58]. Most of the agents available on the market will have the estimated infiltration range described in the flyer. A significant advantage of this technique is its versatility and repeatability, which play a vital role in planning comparable methodologies for clinical trials. However, it seems safe to presume that injections administered outside of MTPs might have a slightly less therapeutic effect. According to our knowledge, there are no clinical trials comparing the effectiveness of both of the above-described methods; therefore, it is hard to objectively assess which approach provides better results.

### 4.4. Techniques Based on Additional Navigation Methods or Support Systems

In the evaluated material, there were few clinical trials using the support methods of localizing the muscles and MTPs, which was quantitatively presented in the results section of the following article.

EMG is one of the most commonly used tools for navigating the intramuscular injections. Ernberg et al. performed the injections, using an audioamplified device to guarantee intramuscular administration [25]. Patel et al. used EMG 27-gauge monopolar electrode needles for administering the botulinum toxin [34], while Sipahi-Calis et al. deposited BTX in line with reflex measurements in electromyography guidelines [13]. All of the abovementioned trials used EMG for assessment of the superficial muscles, masseter and temporalis, apart from Patel’s et al.—they described the access to the LPM due to usage of an EMG electrode needle, which allows for deeper tissues penetration. The authors did not specify the type of EMG appliance used for the procedure. The device that could find its application in this type of treatment is Dantec Clavis, a handheld EMG guidance (Clavis, The Dantec Clavis device, distributed by Natus Medical Incorporated, Pleasanton, CA, United States). This compact apparatus is intended as a stimulator for nerve localization or an aid for guidance of injections into muscles. It provides both EMG and stimulation functions. The application of the device suggested by the producer includes dystonia, strabismus, essential tremor, spasticity and temporomandibular dysfunction [59]. A PubMed database search using the term “Dantec Clavis” does reveal six results although none of the articles describes a case of TMD or myofascial pain treatment. However, the authors claim the use of Dantec Clavis for injecting the muscles, i.e., Wong et al. reported injecting BTX into the muscles with the aid of Dantec Clavis in the management of sixth nerve palsies caused by nasopharyngeal carcinomas [60,61,62,63]. Nevertheless, EMG is a purely functional guidance, and therefore, it does not allow for morphologic tracking [36]. Assuming we would choose the upper head of LPM specifically as the injection target, the EMG might not be the best navigation tool available.

Another approach is to treat the masticatory muscles with percutaneous needle electrolysis (PNE) with the guidance of electrotherapy equipment (EPI ^®^ Advanced Medicine, Barcelona, Spain), simulating the EPI ^®^ technique [44]. PNE is a treatment similar to DDN. In the analyzed paper, the needle used for the procedure was connected to an electrosurgical device, and the EPI ^®^ equipment produced a continuous galvanic current of 2 mA for 3 s in repeated cycles while the needle was inserted into the muscle. According to Lopez-Martos et al., PNE compared to DDN provides significantly earlier improvement with no adverse effects. In 2020, Margalef et al. proposed a study in an animal model investigating the effects of PNE on induced myofascial MTPs [64]. The authors concluded that the higher doses of electrical current are more effective for decreasing MTPs in animal models; therefore, PNE could be more effective than DDN [64].

Another navigation method allowing for proper LPM recognition was described by Pons et al., and it introduces MRI supported planning [36]. The treatment protocol involved conducting a full head MRI of each patient and transferring the scans on the infrared Brainlab software platform (Brainlab AG, Munich, Germany); then, the virtual planning software established the puncture target spot in the central area of the upper head of the LPM as well as the needle’s entry point. In the next stage, patients were situated in the supine position on the operating table, and a headband equipped with reflective markers was placed on their head. Next, they were registered in the navigation system with a contact-free laser point and calibration needle guided by the Brainlab software. Finally, the needle was switched for an intramuscular needle, and the injections were performed following the needle’s real-time progression on the navigation screen. Authors of the trial emphasized the high safety and accuracy of the procedures, although only four out of six treated patients claimed pain improvement [36]. Even though such a small study group does not allow for an objective assessment of the presented method, it nevertheless seems quite complicated, time consuming and costs generating, while most likely achieving comparable results with other, less exacting methods, such as the support of EMG or USG.

Apart from the already investigated approaches, there are more scientific papers describing other alternative methods, such as the use of CT scans as well as USG support; however, those articles do not fully meet our eligibility criteria. Nonetheless, we find it vital for our work to shortly present the techniques applied. Oliveira et al. and Yoshida et al. investigated the technique of intraoral access to the LPM with custom-made guides. Patients’ CT scans were used for virtual modeling with a support of medical imaging software; afterward, customized injection guides were manufactured. The devices obtained in both trials took the form of maxillary splints with a guiding tube for the injection needle on the side [65,66]. They allowed for precise access to LPM and seem to be feasible and reliable tools. Sanabria et al. applied in their study USG imaging, using strain elastography and a novel B-mode spatiotemporal filter in the injection treatment model on porcine masseters. They stated that the proposed method contributes to objectivize ultrasound guided injections as well as to monitor injectate spread within the muscles [67]. The shear wave elastography allows for noninvasive assessment of soft tissue hardness; therefore, it can be a vital tool in localizing the MTPs, which are common target spots for needling therapies. Moreover, elastography is the right tool to evaluate the myofascial pain in correlation with muscles stiffness as well as the treatment results of injections therapies [68,69]. It is a promising tool for the assessment of masticatory muscles, especially the masseter as stated by Olchowy et al. although there still is a significant need for more studies on larger groups to determine the accuracy of elastography to characterize masticatory muscles and their disorders [68].

Eventually, it is worth discussing the masticatory muscles as a primary needling treatment target in all the analyzed clinical trials, as the results indicate that needling therapies are most often carried out within the masseter and temporal muscle. There seem to be few correlated reasons for the following tendency: both the masseter and temporalis are muscles characterized by direct, easy access regarding needling therapies; they have significantly greater volume and are characterized by a greater contraction force than LMP and MPM. Therefore, treatments carried out solely on these muscles tend to bring satisfactory results. Consequently, the costs as well as the time needed to conduct needling interventions within these muscles are lower since usually there is no need for additional guidance systems localizing the muscles. Despite the above mentioned, still there are significant indications for approaching the LPM and MPM as well. Apart from apparent need when LPM and/or MPM are a particular reason for the direct or referred pain with developed MTPs, there also is a diagnostic application for LPM injections as well as the potentiality to treat certain types of TMJ internal derangements [70]. Litko et al., in the study based on MR scans of 191 patients, determined three main patterns of LMP attachment to the TMJ’s disc–condyle complex and stated a positive correlation primarily between anteromedial displacement of the disc and the presence of LMP head inserting solely the TMJ capsule [71]. Importantly, it was highlighted that the attachment type is most likely just a cofactor of the internal derangements and there have to be other factors, such as trauma or a disrupted joint structure, for the displacements to occur [71]. In the cases discussed, application of the botulinum toxin into the LMPs may allow for treatment of the derangements as well as aid in the diagnosis of the acoustic symptoms originating from TMJ, as weakening the LMPs should result in a reduction of the reciprocal click [70,72].

A simple relationship between the needling of the masticatory muscles and the general condition of the patient or the influence on therapies in other areas of dentistry has not been proven. Nevertheless, these relationships should be explored and are potentially possible in the context of both the complications and the potential benefits of needling the masticatory muscles in the course of other treatments. It is known that each tissue disruption, including injections and dry needling of the masticatory muscles, carries the risk of not only bacterial infection, but also viral implantation, including HIV or HCV [73]. However, we have not noted any studies showing an increased infection potential over the course of needling in the masticatory muscles, compared to other procedures involving punctures. Consideration should be given to the possibility of intentionally weakening the activity of the masticatory muscles or eliminating muscle pain in pathologies manifested by abnormal work of the temporomandibular joint, including fractures of the mandible or its defects, e.g., missing teeth [74,75,76]. Of particular importance among mandibular fractures may be intracapsular fractures of the mandibular head, the treatment of which may be highly traumatizing to the structures of the temporomandibular joint, and thus, affect the functioning of the entire stomatognathic system [77,78]. Another issue is the potential of using masticatory muscle needling to stabilize the already achieved therapeutic effect, which may be of importance in orthodontics or dental prosthetics [79,80].

## 5. Conclusions

The interest in needling therapies within masticatory muscles has clearly increased in recent years. Over 85% of articles on this subject were published in the second half of the last decade. The analysis of the literature proves that there is a lack of consistency and standardization in the currently applied treatment methods; therefore, an objective comparison of available clinical trials is very difficult. Our work is a response to the urgent need to systematize knowledge about the dynamically developing field of needling in the masticatory muscles. Due to the focus on needling techniques, this systematic review did not take into account the scientific quality, bias, or conflict of interest of individual qualified studies, which may be a limitation of our paper for some applications. However, the strength of this work is the large number of current clinical trials (n = 28), which were analyzed and summarized.

In the developed literature, more than half of the works describe the needling of the masseter, followed by the temporal muscle, due to achieving better therapeutic results than needling of the lateral and medial pterygoid muscles. Nearly three out of every four papers describe palpation as a method of determining the piercing points. The remaining authors use a variety of navigational techniques, including EMG, EPI, and MRI, which are mostly used as an aid in localizing LPM or MPM. All of the analyzed papers claim the effectiveness of the needling therapies in decreasing pain, increasing maximum mouth opening and/or improving the pain threshold. Therefore, it seems justified to conclude that needling of the masticatory muscles can be a successful supporting and alternative therapy in treatment of myofascial pain.

## Figures and Tables

**Figure 1 ijerph-18-09552-f001:**
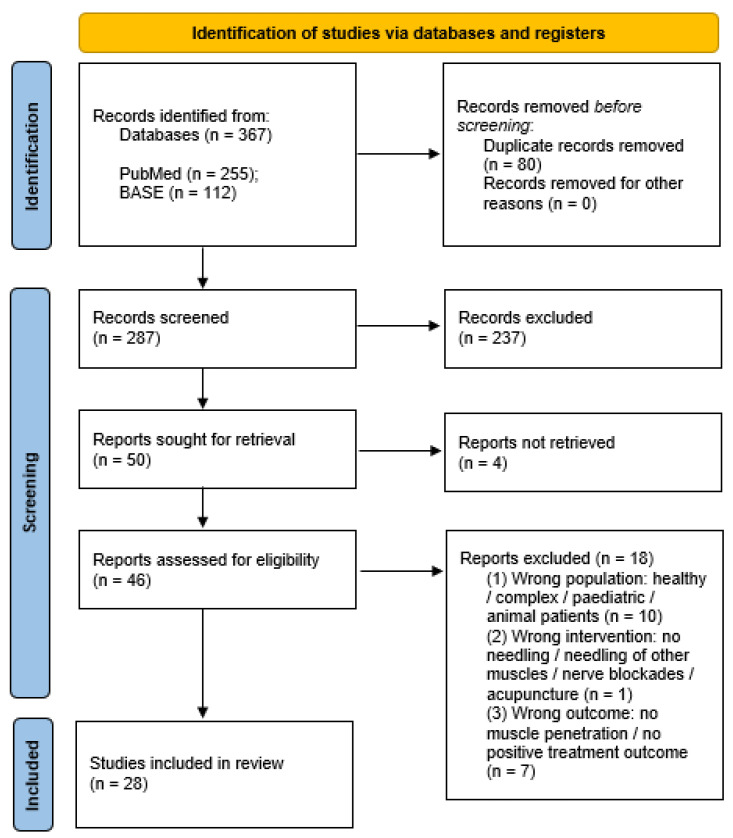
A PRISMA flow diagram of the literature screening and selection processes [19].

**Figure 2 ijerph-18-09552-f002:**
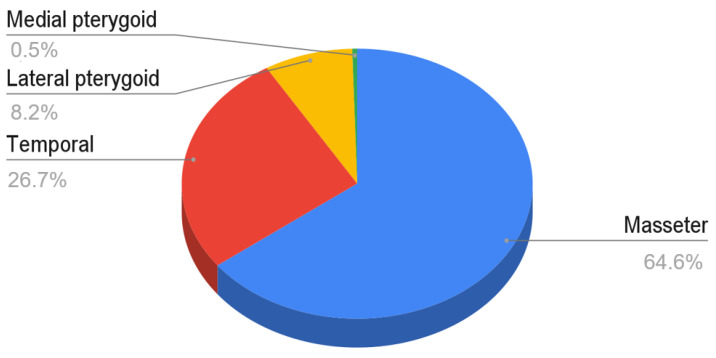
The number of interventions in the form of needling of each of the masticatory muscles in relation to the total number of patients included in the analyzed material.

**Figure 3 ijerph-18-09552-f003:**
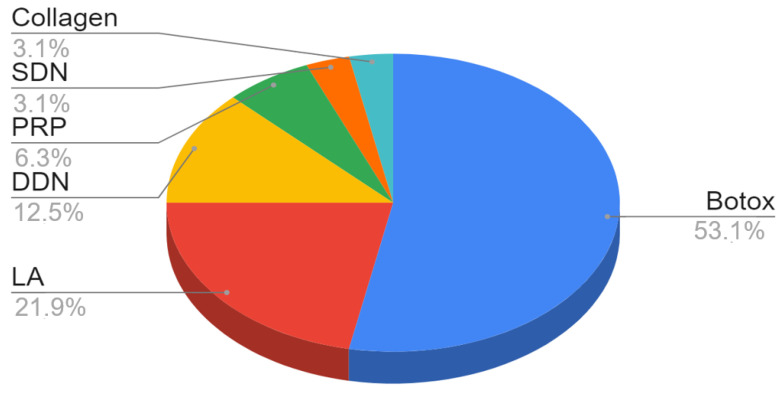
The share of substances injected into the masseter muscles and dry needling in relation to the number of analyzed trials.

**Figure 4 ijerph-18-09552-f004:**
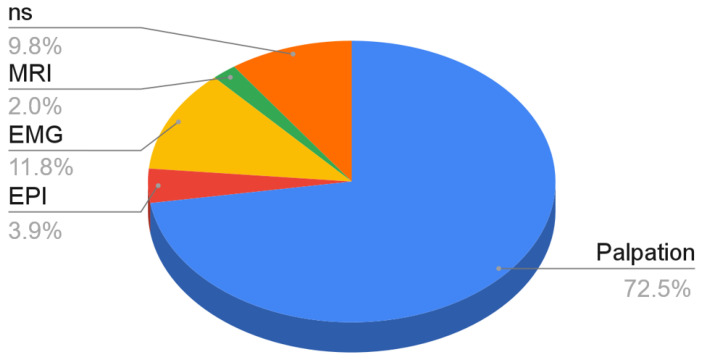
Navigational methods in needling of the masticatory muscles in relation to the number of analyzed studies.

**Table 1 ijerph-18-09552-t001:** Applied search strategies.

	Search Strategy
PubMed	(masseter OR temporalis OR lateral pterygoid OR medial pterygoid OR masticatory OR mastication) AND (muscle OR muscles OR intra-muscular OR intramuscular OR intra muscular OR trigger points) AND (injection OR injections OR puncture OR punctures OR administration OR needling OR dry needling OR acupuncture) AND (pain OR discomfort OR myofascial pain OR antinociceptive OR anti-nociceptive) AND (therapy OR treatment OR management OR pharmacotherapy OR diagnosis OR diagnostic OR diagnostics) AND (technique OR techniques OR method OR methods OR protocol OR protocols OR guidelines)
BASE	(masseter OR temporalis OR pterygoid OR masticatory OR mastication) AND (muscle OR muscles OR intra-muscular OR intramuscular OR intra muscular OR trigger points) AND (injection OR injections OR puncture OR punctures OR administration OR needling OR dry needling OR acupuncture) AND (pain OR discomfort OR myofascial pain OR antinociceptive OR anti-nociceptive) AND (therapy OR treatment OR management OR pharmacotherapy OR diagnosis OR diagnostic OR diagnostics) AND (technique OR techniques OR method OR methods OR protocol OR protocols OR guidelines)

**Table 2 ijerph-18-09552-t002:** The eligibility criteria according to PICOS [16].

	Inclusion Criteria	Exclusion Criteria
Population	Adult patients with myofascial orofacial pain according to the RDC/TMD or DC/TMD or ICOP	Animal patients, patients suffering with fibromyalgia, neuralgia, oromandibular dystonia, cerebral palsy, hypertrophic masseter muscles, clinically induced pain
Intervention	Intramuscular injections into masticatory muscles, dry needling of masticatory muscles	Intramuscular injections into other than masticatory muscles, nerve blockades, acupuncture
Comparison	Any or none	-
Outcome	Primary outcome: Effective penetration of the masticatory muscle in the course of its needling. Secondary outcome: Any positive treatment outcome (eg. pain decrease, increase in maximum mouth opening, pressure pain threshold increase, MTPs desactivation etc.)	-
Study design	Primary studies with an evidence level 1 to 3; description of the needling technique	Papers published prior to 2011; non-English articles

**Table 3 ijerph-18-09552-t003:** Studies regarding masseter muscle injections.

Author, Year of Publication	Study Group Size	Wet/Dry Needling (Injected Substance)	Navigation Method or Support System	Needle Penetration Spot(s)	Amount of Injected Substance (per Muscle)	Number of Treatment Sessions
Al-Wayli, 2017 [21]	25	BTX	manual palpation; deposition spots located regularly in the lower part of the muscle	3 sites	20 IU	1
Ananthan, 2019 [22]	25	LA (2% lidocaine)	digital palpation	MTPs	ns	1
Chaurand, 2020 [23]	22	BTX	digital palpation	2 sites	30 IU	1
Chaurand, 2017 [24]	11	BTX	manual palpation; one spot near mandibular angle and one near zygomatic arch	2 sites	20 IU	1
Ernberg, 2011 [25]	21	BTX	EMG; 15 mm deep	3 sites	50 IU	2
Ghavimi, 2019 [26]	61	BTX	manual palpation; deposition spots located regularly along the long axis of the muscle; 10 mm deep	3 sites	50 IU	1
Guarda-Nardini, 2012 [27]	15	BTX	manual palpation; 2 cm skin surface over most prominent area of the muscle after clenching	minimum 5 sites in reverse pyramid pattern	ns	1
Hosgor, 2020 [28]	44	BTX	manual palpation	3 sites	150 IU	1
Ivask,2016 [29]	20	BTX	manual palpation; most painful spot	5 sites	ns	1
Kang, 2018 [30]	24	LA (1,5 or 5 mg morphine sulfate)	manual palpation; most painful spot	1 site	0.3 mL	1
Kang, 2018 [30]	11	LA (2% lidocaine)	manual palpation; most painful spot	1 site	0.3 mL	1
Kim, 2016 [30]	21	BTX	manual palpation	3 sites	150 IU	1
Meral, 2019 [31]	25	BTX	Ns	6 sites	24 IU	1
Nitecka-Buchta, 2019 [32]	29	PRP	manual palpation; 5–10 mm deep	3 MTPs near the origin, under zygomatic arch	1.5 mL	1
Nitecka-Buchta, 2018 [9]	18	collagen	manual palpation; 10–15 mm deep	MTPs	2 mL	2
Nitecka-Buchta, 2018 [9]	15	LA (2% lidocaine)	manual palpation; 10–15 mm deep	MTPs	2 mL	2
Ozden, 2018 [33]	20	DDN	manual palpation and analogue algometer; 10 mm deep	MTPs	-	3
Ozden, 2018 [33]	20	SDN	manual palpation and analogue algometer; 5 mm deep	MTPs	-	3
Patel, 2017 [34]	20	BTX	EMG and monopolar electrode injection needle	ns	50 IU	1
Pihut, 2016 [35]	42	BTX	manual palpation, area of the greatest cross-section surface of both masseter bellies	ns	21 IU	1
Pons, 2018 [36]	6	BTX	manual palpation	3 sites	30 IU	1
Sipahi Calis, 2019 [13]	9	BTX	EMG	ns	30 IU	1
Taskesen, 2020 [37]	15	DDN	manual palpation	MTPs	-	2
Taskesen, 2020 [37]	15	LA (2% lidocaine)	manual palpation	MTPs	0.2 mL	2
Tesch, 2019 [38]	5	DDN	manual palpation	MTPs	-	3
Uemoto, 2013 [39]	7	LA (2% lidocaine)	digital palpation; 10–20 mm deep	MTPs	0.25 mL	4
Uemoto, 2013 [39]	7	DDN	digital palpation	MTPs	-	4
Villa, 2018 [40]	28	BTX	Ns	3 sites	50 IU	1
Yilmaz, 2020 [41]	26	BTX	manual palpation	MTPs	10 U/MTP	1
Yilmaz, 2020 [41]	27	LA (3% mepivacaine)	manual palpation	MTPs	0.5 mL/MTP	1
Yilmaz, 2020 [41]	29	PRP	manual palpation	MTPs	0.5 mL/MTP	1
Yurttutan, 2019 [42]	48	BTX	manual palpation; deposition spots located regularly across the muscle mass	5 sites	30 IU	1

**Table 4 ijerph-18-09552-t004:** Studies regarding temporal muscle injections.

Author, Year of Publication	Study Group Size	Wet/Dry Needling (Injected Substance)	Navigation Method or Support System	Needle Penetration Spot(s)	Amount of Injected Substance (per Muscle)	Number of Treatment Sessions
Ananthan, 2019 [22]	25	LA (2% lidocaine)	digital palpation	MTPs	ns	1
Chaurand, 2020 [23]	22	BTX	digital palpation	2 sites	20 IU	1
Chaurand, 2017 [23]	11	BTX	digital palpation	1 MTP	10 IU	1
Guarda-Nardini, 2012 [27]	15	BTX	manual palpation; 2 cm skin surface over most prominent area of the muscle after clenching	multiple injections in chess-board pattern	ns	1
Hosgor, 2020 [28]	44	BTX	manual palpation	2 sites	100 IU	1
Ivask, 2016 [29]	20	BTX	manual palpation; most painful spot	5 sites	ns	1
Kim, 2016 [14]	21	BTX	manual palpation	3 sites	100 IU	1
Meral, 2019 [31]	25	BTX	Ns	3 sites	12 IU	1
Patel, 2017 [34]	20	BTX	EMG and monopolar electrode injection needle	ns	25 IU	1
Pons, 2018 [36]	6	BTX	manual palpation	2 sites	20 IU	1
Sipahi Calis, 2019 [13]	9	BTX	EMG	ns	20 IU	1
Villa, 2018 [40]	28	BTX	Ns	2 sites	25 IU	1
Yurttutan, 2019 [42]	48	BTX	Ns	3 sites	15 IU	1

**Table 5 ijerph-18-09552-t005:** Studies regarding lateral pterygoid muscle injections.

Author, Year of Publication	Study Group Size	Wet/Dry Needling (Injected Substance)	Navigation Method or Support System	Needle Penetration Spot(s)	Amount of Injected Substance (per Muscle)	Number of Treatment Sessions
Gonzalez-Perez, 2015 [43]	24	DDN	manual palpation; proper puncture confirmed by jump reaction or local twitch response	MTPs	-	3
Lopez-Martos, 2018 [44]	20	DDN	EPI ^®^ electrotherapy equipment	MTPs	-	3
Lopez-Martos, 2018 [44]	20	PNE	EPI ^®^ electrotherapy equipment	MTPs	-	3
Patel, 2017 [34]	20	BTX	EMG and monopolar electrode injection needle	ns	10 IU	1
Pons, 2018 [36]	6	BTX	MRI infrared navigation; center of the upper head; about 33 mm from the skin	1 site	20 IU	1

**Table 6 ijerph-18-09552-t006:** Studies regarding medial pterygoid muscle injections.

Author, Year of Publication	Study Group Size	Wet/Dry Needling (Injected Substance)	Navigation Method or Support System	Needle Penetration Spot(s)	Amount of Injected Substance (per Muscle)	Number of Treatment Sessions
Mesa-Jimenez, 2020 [45]	5	DDN	needle insertion at the inferior angle of the mandibular bone, parallelly to the mandible, advanced from an inferior to superior direction to maximum 30 mm; confirmed by pain referral during insertion	medial surface of the inferior angle of the mandible	-	1

## Data Availability

The study data may be available on request.

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
