# Peer review of "Intramuscular Injections and Dry Needling within Masticatory Muscles in Management of Myofascial Pain. Systematic Review of Clinical Trials"

_ijerph, 2021, doi:10.3390/ijerph18189552_

Round 1

Reviewer 1 Report

This review is interesting and important for understanding effectiveness of Intramuscular injections and dry needling for myofascial pain. But several points must be improved before publiction. I suggest some revisions in order to improve the manuscript.

- Authors mentioned that this review conducted according to the PRISMA protocols. Thus, protocol of this systematic review should be presented in "Method section". (pg 3, line 97)

e.g. https://doi.org/10.3390/jcm10143087

- Authors presented the PICO and search strategy. (pg 4, line 151-154) First, outcome measure is not clear. what is primary outcome or secondary outcome? It is very important for concluding effectiveness of systematic review. Second, keyword "protocol" is not proper for systematic review.

- I recommend authors to re-arrange "Materials and Methods section" following sub-title (pg 3-4, line 97-134); 2.1. Prototol registration, 2.2. Data sources and Search strategy, 2.3. Eligibility Criteria, 2.4. Study Selection, 2.5. Data Extraction, 2.6. Risk of Bias assessment (if RCTs is included in this review)

- I recommend authors to present flowchart(table 3) using PRISMA flow diagram (refer https://doi.org/10.3390/jcm10143087). And I think that it is not table but figure. (pg 5, line 155-156)

- In table 3, "wrong population, wrong type of article, background articles, wrong intervetion, and wrong outcome" should be presented in more details. (pg 5, line 155-156)

e.g) wrong population -> healty population or patient with no myofascial pain

e.g.) wrong type of article -> not clinical studies

e.g.) wrong intervention -> intervention is not Intramuscular injections and dry needling

- In table 4, informations about comparision, outcome measure and main result of outcome measure are not reported. And I recommend authors to supplement information of study design of clinical studies (e.g. RCT, CCT, cohort study, or case study)

Author Response

Reviewer 1

This review is interesting and important for understanding effectiveness of Intramuscular injections and dry needling for myofascial pain. But several points must be improved before publiction. I suggest some revisions in order to improve the manuscript.

Thank you for reading our work and for critically evaluating it. We are very pleased that the Reviewer found the paper valuable and interesting. We appreciate the very detailed and extremely pertinent comments that allowed us to improve our manuscript.

- Authors mentioned that this review conducted according to the PRISMA protocols. Thus, protocol of this systematic review should be presented in "Method section". (pg 3, line 97)

e.g. https://doi.org/10.3390/jcm10143087

As suggested, we have rearranged and improved the "Method section". Following the example, we have divided into subsections: 2.1. Protocol registration; 2.2. Data sources and Search strategy; 2.3. Eligibility Criteria; 2.4. Study Selection; 2.5. Data Extraction. The applied corrections were made in the manuscript text.

Protocol registration

The protocol of this systematic review was developed on April 6, 2021 and submitted to the scientific council of the Department of Temporomandibular Disorders in Zabrze, Medical University of Silesia in Katowice. This systematic review was conducted in accordance with the Preferred Reporting Items for Systematic Reviews and Meta-Analyzes (PRISMA) statement (Supplemental 1). In the course of the systematic review, eligibility criteria were established and respected using the PICOS framework [16].

Supplemental 1: PRISMA 2020 Checklist

Data sources and Search strategy

The MEDLINE and PubMed Central (PMC) as well as other databases including journal archives were searched using PubMed search engine and BASE i.e. Bielefeld Academic Search Engine on April 18, 2021 [17, 18]. At the time of the search, PubMed engine serviced databases with a total of over 32 million records, and BASE - databases with over 240 million records [17, 18]. The structure of each search strategy is presented in Table 2.

Eligibility Criteria

Eligibility criteria for this systematic review have been developed according to the PICOS framework [16]. These were used to develop a search strategy in medical databases of records and to the subsequent stages of article selection. In line with the PICOS acronym, positive (inclusion) and negative (exclusion) criteria were adopted. (1) In line with the “Population” trait, studies on groups of adult patients suffering from myofascial pain in the mouth and / or face were included. Studies on animals and studies on human patients suffering from specific diseases and syndromes listed in the appropriate cell of Table 1 were excluded. Studies in which pain was induced by researchers were also rejected, due to additional injection necessary for that purpose. (2) Therapeutic punctures into the masticatory muscles, including both dry needling and the administration of any substances, were considered necessary for the qualification of the article in the context of “Intervention”. However, punctures which, apart from the masticatory muscles, also affected other muscles, served nerve blocks or served acupuncture, were rejected. (3) No “Comparison” was required, i.e. research work was accepted, regardless of the type of control group or its absence. (4) With regard to “Outcome”, the description of the puncture technique allowing reaching the interior of any masticatory muscle was adopted as a necessary condition. Additionally, it was required to indicate in the article any of the positive puncture effects mentioned in the appropriate cell of Table 1. (5) As a positive criterion "Study design" clinical studies were required with an evidence level of 1 to 3, i.e. randomized controlled trials, prospective cohort studies and retrospective case control studies were accepted. Articles published before 2011 and not available in English were rejected. The full PICOS criteria, i.e. with additional details, are presented in Table 1 [16].

Study Selection

Following systematic review was conducted based on the Preferred Reporting Items for Systematic Reviews and Meta - Analyses (PRISMA) guidelines with the workflow presented in Figure 1 [19]. Initially all identified records were synthesized using the Rayyan QCRI application [20], the duplicates were removed and afterwards titles and abstracts were blindly screened by two of the authors (Z.N. and M.C.) according to PICOS criteria. Cohen’s Kappa coefficient was used to determine the convergence of the assessments in the screening phase and the result was κ = 0,98. Each time a decision conflict whether to include or exclude an article occurred the abstract was qualified to the next evaluation level - full-text assessment. Afterwards three of the authors (Z.N., M.C. and A.N.B.) blindly evaluated the full-texts reaching total convergence of the results.

Data Extraction

The data was extracted by three authors (Z.N., M.C. and A.N.B.), who obtained the following information from the content of the articles: (1) Author, year of publication; (2) Study group size; (3) Wet / dry needling

(injected substance); (4) Navigation method or support system; (5) Needle penetration spot(s); (6) Amount of injected substance (per muscle); (7) Number of treatment sessions. These data have been compiled in a table using the Google Sheets software (Google LLC, Mountain View, California, United States).

- Authors presented the PICO and search strategy. (pg 4, line 151-154) First, outcome measure is not clear. what is primary outcome or secondary outcome? It is very important for concluding effectiveness of systematic review. Second, keyword "protocol" is not proper for systematic review.

We fully agree with the opinion that the outcomes have not been defined clearly enough. We propose a revised version of this and “Study design” rows of the PICOS table.

Outcomes:

Inclusion criteria:

Primary outcome: Effective penetration of the masticatory muscle in the course of its needling.

Secondary outcome: Any positive treatment outcome (eg. pain decrease, increase in maximum mouth opening, pressure pain threshold increase, MTPs desactivation etc.)

Exclusion criteria: -

Study design:

Inclusion criteria: Primary studies with an evidence level 1 to 3; description of the needling technique

Exclusion criteria: Papers published prior to 2011; non-English articles

- I recommend authors to re-arrange "Materials and Methods section" following sub-title (pg 3-4, line 97-134); 2.1. Prototol registration, 2.2. Data sources and Search strategy, 2.3. Eligibility Criteria, 2.4. Study Selection, 2.5. Data Extraction, 2.6. Risk of Bias assessment (if RCTs is included in this review)

We distinguished the sections as suggested. Thank you very much for the proposed amendment, which allowed for a clearer presentation of the content.

- I recommend authors to present flowchart(table 3) using PRISMA flow diagram (refer https://doi.org/10.3390/jcm10143087). And I think that it is not table but figure. (pg 5, line 155-156)

Thank you for paying attention to the graphic design of the PRISMA flowchart and its incorrect labeling as a table. The person responsible for the illustrations in our team has already corrected the figure.

- In table 3, "wrong population, wrong type of article, background articles, wrong intervetion, and wrong outcome" should be presented in more details. (pg 5, line 155-156)

e.g) wrong population -> healty population or patient with no myofascial pain

e.g.) wrong type of article -> not clinical studies

e.g.) wrong intervention -> intervention is not Intramuscular injections and dry needling

Indeed, the mere reference to the adopted eligibility criteria is unclear. We improved the content of the flowchart description.

Figure 1. PRISMA flowchart. Detailed exclusion criteria: (1) Wrong population: healthy / complex / pediatric / animal patients; (2) Wrong intervention: no needling / needling of other muscles / nerve blockades / acupuncture; (3) Wrong outcome: no muscle penetration / no positive treatment outcome

- In table 4, informations about comparision, outcome measure and main result of outcome measure are not reported. And I recommend authors to supplement information of study design of clinical studies (e.g. RCT, CCT, cohort study, or case study)

As the purpose of this systematic review was not to assess the effectiveness of different treatment techniques by different teams, but to compile muscle needling techniques, we considered the inclusion of data from the control group and any evaluation of treatment outcomes unreasonable. Similarly, the type of study in the context of a detailed assessment of its degree of evidence does not apply here, because we are interested in the fact of describing a given needle technique and a zero-one assessment of the sense of its clinical use, taking into account any therapeutic effectiveness (without a qualitative assessment). Therefore, in this systematic review, we did not plan and do not intend to extract the data discussed above. Nevertheless, we thank you for pointing this direction, as we believe that it is a valuable idea for another paper.

Reviewer 2 Report

The manuscript needs to be rewrite. The authors reported that followed PICO Strategy and PRISMA Statement, that are golden standard in Systematic review; but the information described in the manuscript does not follow this documents.

Author Response

The manuscript needs to be rewrite. The authors reported that followed PICO Strategy and PRISMA Statement, that are golden standard in Systematic review; but the information described in the manuscript does not follow this documents.

Thank you for reading our manuscript and expressing your critical opinion. At first, we were disgusted with the laconic nature of this review, the more so as we had already published a number of systematic reviews in MDPI journals and became convinced that our work was valuable. Nevertheless, we undertook a thorough analysis of our manuscript in relation to the said documents, and we must admit that in fact it was necessary to rewrite the manuscript, which we did. Due to the numerous changes and the desire to clearly present them, we allow ourselves not to mention them here, but to attach a new version of the manuscript. Once again, we thank you for your opinion and admit that it was a valuable contribution to reflection on our work.

Round 2

Reviewer 1 Report

The authors have suscessfully addressed all my comments and suggestions.

Author Response

Thank You for all Your valuable comments, the final version of the manuscript is attached to this note

Reviewer 2 Report

The authors improved the quality of the study, but it is necessary to improve the information about limitations and strength of this study and the conclusion.

Author Response

Thank you for Your valuable comments, it has been corrected in the text of the manuscript.
